# The *in vitro* assessment of rheological properties and dentin remineralization of saliva substitutes containing propolis and aloe vera extracts

**Surapong Srisomboon**[1¤a], **Thanapong Intharah**[2], **Ungkarn Jarujareet**[3], **Arnit Toneluck**[1], **Piyaphong Panpisut**[1¤b]*

1 Faculty of Dentistry, Thammasat University, Pathum Thani, Thailand, 2 Visual Intelligence Laboratory, Department of Statistics, Faculty of Science, Khon Kaen University, Khon Kaen, Thailand, 3 NECTEC, National Science and Technology Development Agency (NSTDA), Pathum Thani, Thailand

¤a Current address: Department of Oral Health, Lamlukka Hospital, Pathum Thani, Thailand
¤b Current address: Thammasat University Research Unit in Dental and Bone Substitute Biomaterials, Thammasat University, Pathum Thani, Thailand
* panpisut@tu.ac.th

**Data Availability Statement:** All relevant data are within the manuscript and its Supporting information files.

## Abstract

Saliva substitutes with enhanced dentin remineralization properties were expected to help manage caries progression in patients with xerostomia. This *in vitro* study examined the rheological properties and remineralization action of experimental saliva substitutes containing propolis extract and aloe vera extract on demineralized dentin. Four experimental saliva substitutes were formulated with varying concentrations of propolis extract (P) and aloe vera extract (A) were prepared. A commercial saliva substitute (Biotene Oral Rinse) was used as a commercial comparison. The rheological properties and viscosity of these materials were measured using a strain-controlled rheometer (n = 3). The remineralizing actions of saliva substitutes on demineralized dentin after 2 weeks were determined using ATR-FTIR and SEM-EDX (n = 8). The results were expressed as a percentage increase in the mineral-to-matrix ratio. Biotene demonstrated a significantly higher viscosity (13.5 mPa·s) than experimental saliva substitutes ($p<0.05$). The addition of extracts increased the viscosity of the saliva substitutes from 4.7 mPa·s to 5.2 mPa·s. All formulations showed minimal shear thinning behavior, which was the viscoelastic properties of natural saliva. The formulation containing 5 wt% of propolis exhibited the highest increase in the median mineral-to-matrix ratio (25.48%). The SEM-EDX analysis revealed substantial mineral precipitation in demineralized dentin, especially in formulations with 5 wt% or 2.5 wt% of propolis. The effect of the aloe vera extract was minimal. The addition of propolis and aloe vera extracts increased the viscosity of saliva substitutes. the addition of propolis for 2.5 or 5 wt% to saliva substitutes increased mineral apatite precipitation and tubule occlusion. To conclude, the saliva substitute containing propolis extract demonstrated superior remineralizing actions compared with those containing only aloe vera extract.

**Funding:** National Science, Research and Innovation Fund (NSRF) via the Program Management Unit for Human Resources and Institutional Development, Research and Innovation (Grant number B05F650016) The funders had no role in study design, data collection and analysis, decision to publish, or preparation of the manuscript.

**Competing interests:** The authors have declared that no competing interests exist.

## Introduction

It is estimated that by 2050, the proportion of older adults aged over 65 years will increase from 12% to 22% [1]. Furthermore, 80% of older adults will reside in middle-to-low-income countries [1]. Dental caries, particularly root caries, are a significant oral health issue in this demographic. The reported prevalence of root caries varies among continents from 18% to 95%, with the prevalence in Asia and Africa being greater than 50% [2]. This increase in the prevalence is expected to become a considerable health burden in low and middle-income countries [3]. The progression and severity of the caries lesions are attributed to an imbalance of mineral loss (demineralization) and gain (remineralization) [4], which is associated with dental biofilm dysbiosis [5], reduced saliva functions, and high sugar intake [6].

Xerostomia refers to the subjective feeling of dry mouth associated with the loss of salivary functions [7], which can be induced by various medications [8], such as antidepressants, anti-hypertensives, and proton pump inhibitors. Furthermore, using multiple medications (poly-pharmacy) also worsens xerostomia due to a synergistic effect [9]. Radiation or chemotherapy treatments can also cause xerostomia due to salivary gland hypofunction. This could lead to severe impairment of oral functions and quality of life, resulting in a deterioration in overall health due to restrictions in daily activities, social disabilities, and malnutrition [10, 11].

Strategies for maintaining oral functions include enhancing salivary flow through medication or the daily use of saliva substitutes [12]. Key characteristics of an effective saliva substitute include a suitable pH (~7), ion concentration (calcium and phosphate ions), and rheological properties that mimic natural saliva. A systematic review indicated that various commonly commercial products, such as Biotene Oral Balance (GSK, GSK Consumer Healthcare, Warren, NJ, USA), BioXtra Drymouth Systems (BioXtra, Herts, UK), and Oral Balance Gel (GSK, GSK Consumer Healthcare, Warren, NJ, USA), were effective in alleviating symptoms associated with xerostomia in patients suffering from radiation-induced dry mouth [13]. The saliva substitute was also expected to help promote suitable pH and ions that allow mineral precipitation. This may facilitate remineralization, helping to control active caries or encourage tubule occlusion, thereby protecting the pulp from irritation and reducing dentin hypersensitivity [14, 15].

Propolis extract is a resinous substance that may contain different components depending on the geographic area, weather, botanical sources, and type of honeybee [16]. However, it typically consists of 30% wax, 50% resins and plant balsams, 10% volatile oils, 5% dander, and other organic compounds [16]. Several compounds (>300 substances) were isolated and identified, but the most commonly found are phenolic substances, especially flavonoids, belonging to different sub-classes such as flavanones, flavones, flavanols and dihydro flavonols [17]. These flavonoids may provide therapeutic properties, such as antimicrobial activities, anti-inflammation effects, antioxidant properties, and wound healing [18]. Additionally, propolis is a rich source of elements such as magnesium, nickel, calcium, silver, and copper [17, 19]. These trace elements in the resinous components may help promote ion retention and accumulation on the tooth surface, which may enhance the mineralization of the demineralized tooth structure [20]. Propolis also enhanced the ion release and promoted the adherence of calcium phosphate to tooth structure [16].

A previous study examined the effect of chewing gum incorporated with propolis on dentin mineralization [21]. The study also showed that the addition of propolis promoted dentinal tubule occlusion and increased surface microhardness. Another alternative natural extract for saliva substitutes is aloe vera extract. The major components of aloe vera include polysaccharides, organic acids, sugar, phenolic compounds, and minerals [22]. These compounds may provide additional beneficial effects such as moisturizing effect, anti-inflammatory properties,

and wound healing for dried mucosa in xerostomia patients [23]. A study [24] that assessed the remineralizing effect of aloe vera gel/toothpaste showed that aloe vera may help promote mineral precipitation due to its containing calcium and phosphate and a suitable pH for the environment. The aloe vera gel combined with fluoride toothpaste also promoted the increase in surface microhardness of the demineralized enamel [25].

It was highlighted that the differences in viscosity, lubrication, and wetting properties among saliva substitutes can significantly impact oral functions [9, 26]. The saliva substitutes should also replicate the non-Newtonian and viscoelastic nature of natural saliva, changing its viscosity upon different shear rates [27]. The rheological properties of saliva substitutes are commonly assessed using a rheometer [28], but this macro-rheology equipment is considerably expensive and might not be readily available in oral health care centers.

The rheometer was used to determine the bulk or macro-rheological properties of fluids by imposing a shear stress or strain on a measuring sample using rotating cylinders, cones, or plates. Then, the results of the responses due to the rotating probe, such as torque or shear rate, are measured. However, this approach requires milliliter sample volumes for a measurement to obtain a reliable result [29]. In addition, rotational symmetry, overfill and underfill samples, and the liquid-air interface are also challenges in obtaining reliable responses because these are the causes of surface tension that produces torque at a low shear rate [30]. In contrast to bulk rheology measurement, a previous study demonstrated the feasibility of using our Compact Platform for Micro-Rheological Assessment of Micro-Volume Fluids (compact-DDM) method to analyze various body fluids, such saliva [31]. The study also revealed that this technique involved tracing the movement of microparticles in the viscoelastic fluid under a microscope and then inferring the rheological properties of the fluid using the generalized Stokes-Einstein relation [32]. Consequently, the compact-DDM method facilitates the investigation of local rheological properties. In addition, its minimal sample volume requirement (microliters) makes it particularly advantageous for characterizing expensive or difficult-to-obtain materials. Furthermore, the sample is enclosed in an in-house sample container using spacer and glass slides. Therefore, surface tension due to the liquid-air interface is irrelevant in our compact-DDM measurement. These capabilities could be useful for the understanding and precise design of new materials under a specific mechanical response.

The objective of this study was to assess the rheological properties and the *in vitro* remineralization potential of demineralized dentin by experimental saliva substitutes containing propolis and aloe vera extracts. The null hypothesis of the study was that the experimental saliva substitutes showed no significant differences in rheological properties and dentin remineralization compared with the commercial product. The secondary objective was to additionally compare the rheological properties of the experimental saliva substitutes determined by both a strain-controlled rheometer and the compact-DDM. It was expected that the value obtained from both devices should be comparable.

## Materials and methods

### Preparation of artificial saliva

The saliva substitute was performed using the protocol in the previous studies [33–35]. Briefly, all chemicals (Table 1) were weighed using a 4-digit analytical balance (MSDNY-43, METTLER TOLEDO, Columbus, OH, USA) and gradually added to a bottle containing 1000 mL of deionized water. The solution was mixed using a magnetic stirrer at room temperature for ~6 h to ensure the complete dissolution of sodium carboxymethyl cellulose. The final pH was then adjusted to 6.75 using KOH.

**Table 1. T0068e composition of base saliva substitute in the current study.**

| Chemicals | Amount (g/L) | Lot number | Suppliers |
|---|---|---|---|
| Methyl-p-hydroxybenzoate | 2.0 | BCBV5508 | Sigma Aldrich (St. Louis, MO, USA) |
| KCl | 0.625 | BCBW2329 | |
| $MgCl_2 \cdot 6H_2O$ | 0.059 | BCBW0417 | |
| $CaCl_2 \cdot 2H_2O$ | 0.166 | BCBS6619V | |
| $K_2HPO_4$ | 0.804 | SLBZ1598 | |
| $KH_2PO_4$ | 0.326 | SLCD1991 | |
| Sodium Carboxymethyl Cellulose | 10.0 | MKCK6688 | |

Composition and amount of each chemical for preparing saliva substitute for 1000 mL.

**Table 2. The formulation of experimental materials.**

| Formulations | Abbreviation | Propolis extract (wt%) | Aloe vera extract (wt%) | pH (25 ˚C) | Ca (ppm) | P (ppm) |
|---|---|---|---|---|---|---|
| Group 1 | P5A0 | 5 | 0 | 6.72 | 40.4 | 185.9 |
| Group 2 | P0A5 | 0 | 5 | 5.84 | 66.2 | 190.0 |
| Group 3 | P2.5A2.5 | 2.5 | 2.5 | 6.34 | 57.2 | 189.4 |
| Group 4 | P0A0 | 0 | 0 | 6.88 | 45.6 | 209.1 |
| Group 5 | Biotene | - | - | 6.49 | 44.6 | 372.1 |
| Group 6 | DI | Deionized water (control group) | | | | |

The formulation and composition of the extract are in wt%.

Four experimental saliva substitute formulations (Table 2) containing propolis extract (P; Nature Answer, Saratoga Springs, NY, USA) and aloe vera extract (A; Herbal Answer, Hauppauge, NY, USA) were prepared. The commercial material is Biotene Dry Mouth Oral Rinse (GSK Consumer Healthcare, Warren, NJ, USA)(Table 3). Deionized water was used as the negative control. The pH of the material was measured using a pH meter. The ion concentration (Ca, P) was determined using an inductively coupled plasma optical emission spectrometry (ICP-OES, Optima 8300, PerkinElmer, Waltham, MA, USA) [36].

## Assessment of rheological properties

The test was performed using a strain-controlled rheometer (Thermo Scientific HAAKE MARS 40/60 Rheometers, Thermo Fisher Scientific, Waltham, MA, USA). A stainless-steel parallel plate geometry (60 mm in diameter) was attached and used to evaluate the rheological responses of the specimen. Then, an approximately 2 mL sample volume was placed between

**Table 3. The composition of the commercial saliva substitute in the current study.**

| Material | Composition | Supplier |
|---|---|---|
| Biotene Dry Mouth Oral Rinse (Biotene) | Water, glycerin, xylitol, sorbitol, propylene glycol, poloxamer 407, sodium benzoate, hydroxyethyl cellulose, methyl paraben, propylparaben, flavor, sodium phosphate, disodium phosphate | GSK Consumer Healthcare, Warren, NJ, USA |

The composition and chemicals of commercial saliva substitutes from the safety data sheet.

the measuring geometry. The gap between the plate geometry was set at 0.5 mm. The viscosity of the materials (n = 3) was assessed using the oscillatory shear rate from 0.1 to 100 s$^{-1}$.

Additionally, the rheological properties of the saliva substitutes (n = 3) obtained from the rheometer were compared using the Compact Platform for Micro-Rheological Assessment of Micro-Volume Fluids (compact-DDM) [31]. The measurement was performed by tracing the motion of the polystyrene microparticles (1.0 μm in diameter) while moving in the saliva substitutes. The mean squared displacement of the traced microparticles was calculated, and the generalized Stokes-Einstein relation (GSER) was used to obtain the fluid viscoelastic modulus [37]. Accordingly, the corresponding complex viscosity was determined from the moduli. The Cox-Merz rule was then applied to the complex viscosity to estimate the shear viscosity of the saliva substitutes [31]. The temperature during the measurement was controlled at ~25˚C to allow a direct comparison of the results from the rheometer and compact-DDM.

## Assessment of remineralizing actions

Forty-eight extracted human premolars of comparable size and without visible carious and non-caries lesions were collected from the Faculty of Dentistry, Thammasat University, Pathum Thani, Thailand. This study received ethical approval from the Human Research Ethics Committee of Thammasat University (Science), Thailand (approval number 032/2566). The consent for collecting the teeth was waived by the ethics committee because patient identification was not required. The teeth were stored in 0.1% thymol solution at room temperature. The duration for access to collected samples was from 19/12/2023 to 03/01/2024.

The crown of the teeth was cut horizontally sectioned using a diamond cutting machine (Accutom 50, Struers, Cleveland, OH, USA) to obtain a dentin slice with a thickness of approximately 2 mm thick (Fig 1). The specimens were then cleaned in an ultrasonic bath for 5 min. The dentin specimens (n = 8 per group, 6 groups) were demineralized using 17% ethylenediaminetetraacetic acid (EDTA) for 6 h [38]. Subsequently, attenuated total reflectance Fourier transform infrared (ATR-FTIR) spectroscopy (Nicolet iS5, Thermo Fisher Scientific, Waltham, MA, USA) was employed to obtain the FTIR spectra. The specimens were blotted dry and placed on the ATR diamond. The spectra at 700–4000 cm$^{-1}$ were recorded using the resolution of 4 cm$^{-1}$ and 32 scans [39].

The pH cycling protocol was performed to simulate demineralization/remineralization conditions in the oral cavity [40]. Firstly, the demineralized dentin specimens (n = 8 per group, 6 groups) were treated with 100 μL of toothpaste slurry for 2 min. The slurry was prepared at a 1:3 wt/vol ratio using 1.1% NaF fluoride toothpaste (Neutrafluor 5000 Sensitive, Colgate-Palmolive, Sydney, Australia) mixed deionized water, and was freshly made prior to each application. Then, the specimens were blotted dry and placed in a 12-well plate containing 3 mL of demineralized solution composed of 0.4723 g/L Ca(NO$_3$)$_2$·4H$_2$O, 0.2722 g/L KH$_2$PO$_4$, and 75 mmol/L acetic acid, adjusted to a pH of 4.5, for 6 h at 37ºC. Following demineralization, the specimens were repeatedly treated with 100 μL of the toothpaste slurry for 2 min. They were then placed in the 12-well plate containing saliva substitutes or deionized water (groups 1 to 6) for 14 h at 37ºC. This pH cycling protocol was repeated daily for 14 days.

At 7 days and 14 days, the specimens were blotted dry and placed on the ATR diamond. The FTIR spectra at 700 to 4000 cm$^{-1}$ were recorded at a resolution of 4 cm$^{-1}$ and 32 scans each. Then, they were placed in the storage solution. The mineral-to-collagen ratio was determined by dividing the FTIR peak height at ~1024–1030 cm$^{-1}$ of the phosphate group of hydroxyapatite [41] by the peak height at ~1636 cm$^{-1}$ (amide I of collagen) [42] (Fig 2). The percentage increase in this mineral-to-collagen ratio (Abs$_{1024}$/Abs$_{1636}$) at 7 days and 14 days was calculated.

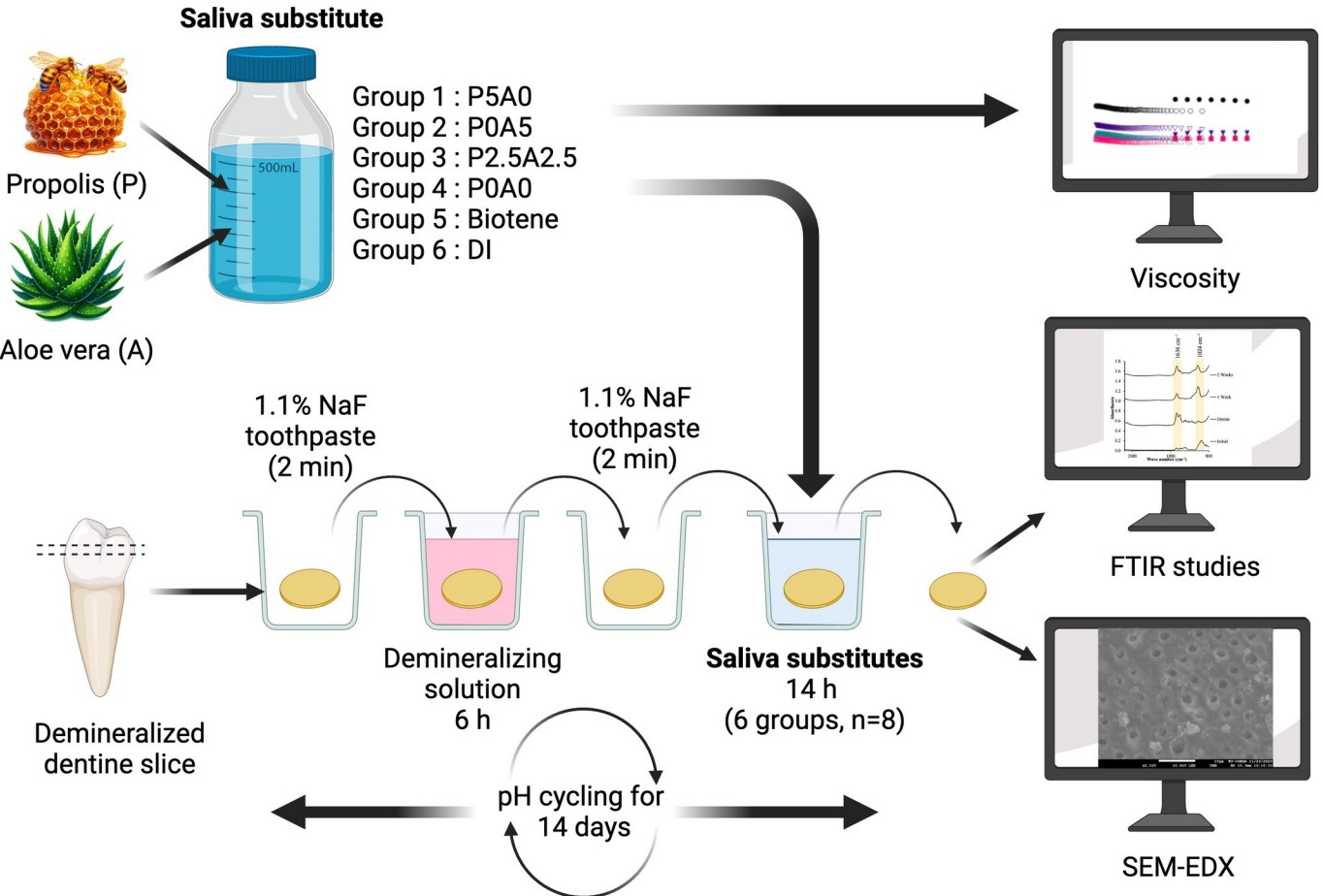

**Fig 1. Protocol for testing in this study.** This figure illustrates the experimental protocol and pH cycling step for testing remineralization used in this study. Created with BioRender.com.

## Assessment of surface mineral precipitation

At 2 weeks, a representative specimen was randomly selected and blotted dry. The specimen was sputter coated with Au using a sputter coating machine (Q150R ES, Quorum Technologies, East Sussex, UK) at a current of 23 mA for 45 s. The surface of the specimen was examined using a scanning electron microscope (SEM, JSM 7800F, JEOL Ltd., Tokyo, Japan) at an accelerating voltage of 10 kV, a working distance of 10.3 mm, with magnifications ranging from 2,500× to 10,000×. Elemental analysis of the precipitates was conducted using a dispersive X-ray spectrometer (EDX, X-Max 20, Oxford Instruments, Abingdon, UK) to analyze the elemental composition of the precipitate on representative specimens. The measurement was taken from three different areas of the precipitate with a magnification of 5000x and a beam voltage set at 10 kV. Data were analyzed using INCA software version 5.05 (ETAS, Stuttgart, Germany).

## Statistical analysis

Results were present as mean ± standard deviation (SD) for normally distributed data and as median with minimum-maximum range (min-max) for non-normally distributed data. Raw

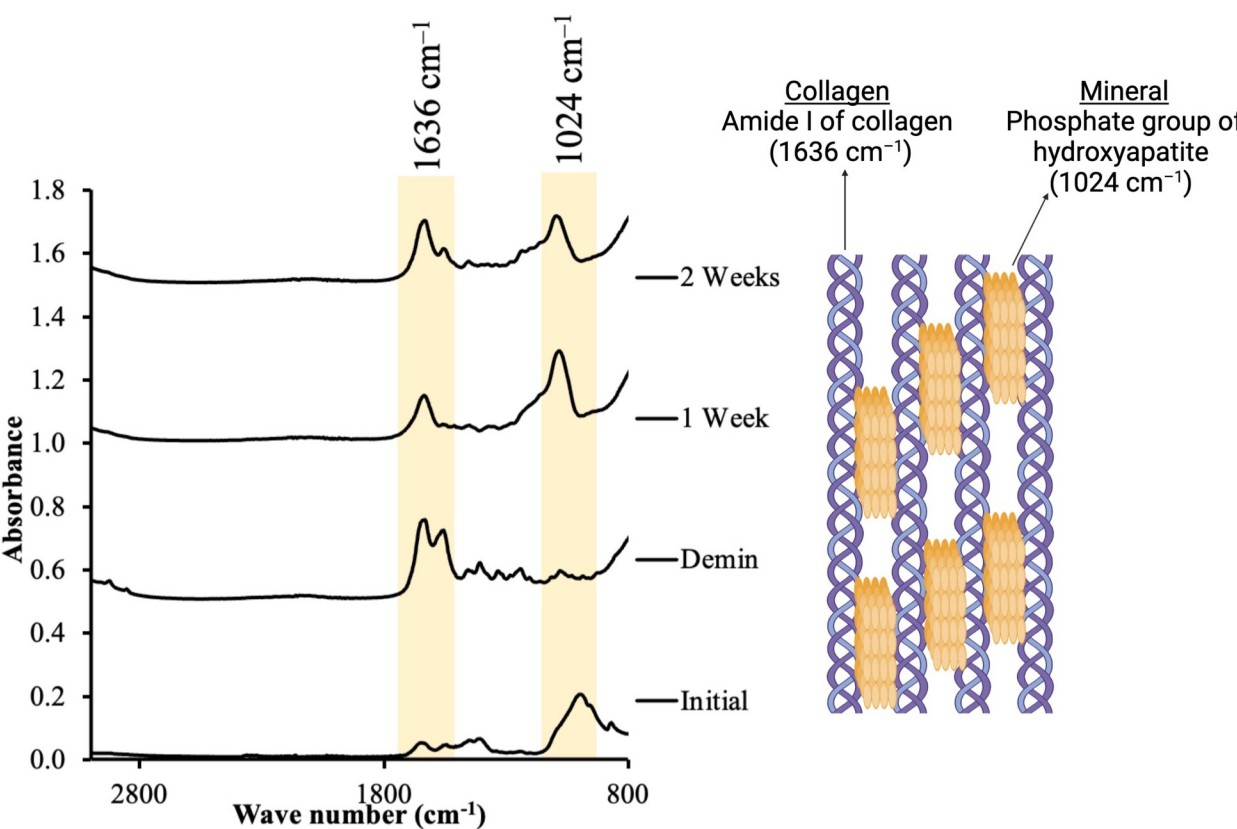

**Fig 2. The example of FTIR result.** The height at ~1636 cm$^{-1}$ and ~1030–1024 cm$^{-1}$ represent asymmetric stretching vibration of the phosphate group of hydroxyapatites and stretching vibration of the peptide carbonyl group (–C = O) of amide I from collagen. Created with BioRender.com.

data are also provided in S1 File. Data were analyzed using Prism version 10.1.0 for macOS (GraphPad Software, San Diego, CA, USA). The distribution of data was determined using the Shapiro-Wilk test. For rheological properties, one-way ANOVA followed by Tukey's post hoc multiple comparison test was performed. The Kruskal-Wallis test, followed by the Dunn procedure, was used for the remineralization test. The Wilcoxon matched-pairs signed rank test was also used to compare the mineral-to-collagen ratio at 1 week and 2 weeks within the same material. A significance level was set at $p$ = 0.05. Additionally, the power analysis conducted using G*power version 3.1 (University of Dusseldorf, Dusseldorf, Germany) [43], based on effect sizes and variability observed in previous studies [39, 44], indicated that sample size in each group should provide power greater than 0.95 at an alpha level of 0.05.

## Results

### Assessing rheological properties

The result of shear sweep measurements indicated a slight decrease in viscosity with increasing shear rate, specifically in groups 2 and 5 (Fig 3A). The highest viscosity was detected with Biotene (Fig 3B). The average viscosity at the shear rate of 10 to 100 s$^{-1}$ obtained from Biotene (13.45±0.03 mPa·s) was also significantly higher than other materials ($p$<0.01). For the experimental materials, group 1 (5.24±0.05 mPa·s) demonstrated the highest viscosity, which was

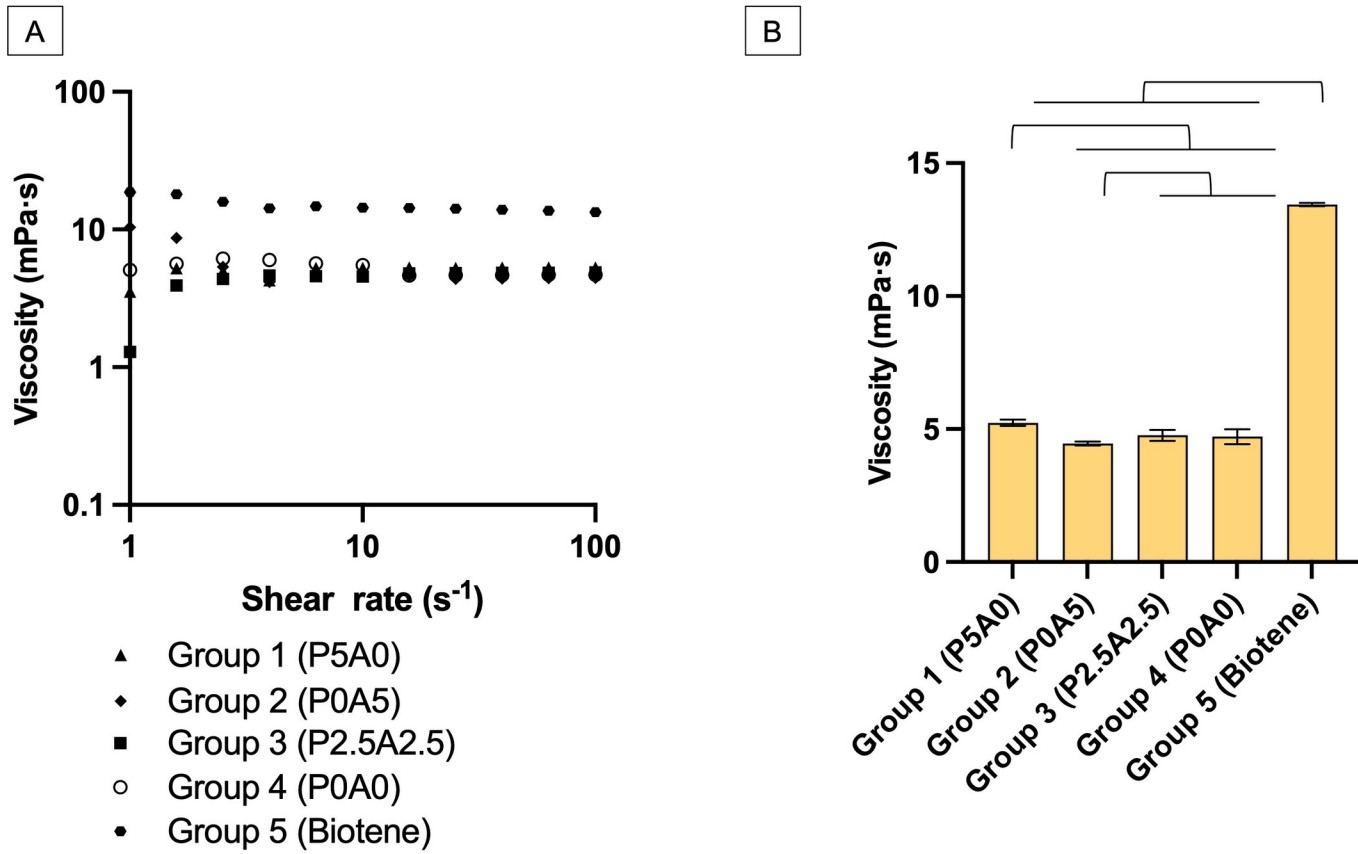

**Fig 3. Results from the strain-controlled rheometer.** (A) The viscosity upon changing the shear rate of each material. (B) The average viscosity of materials at a shear rate of 10 to 100 $s^{-1}$. Error bars are SD (n = 3). Lines indicate $p < 0.05$.

significantly higher than that of group 3 (4.77±0.09 mPa·s), group 4 (4.71±0.13 mPa·s) and group 2 (4.46±0.03 mPa·s)($p < 0.05$).

Our compact-DDM device was unable to obtain the viscosity at a high shear rate due to the limit of the camera frame rate employed. Therefore, the results from the overlapped shear rate region between both techniques (6.38 to 17.65 $s^{-1}$) were compared (Table 4). A similar trend of data was detected where Biotene showed much higher viscosity than the experimental materials. The results obtained from the DDM technique were significantly different from those

**Table 4. Result from strain-controlled rheometer and the compact-DDM.**

| Formulations | Rheometer | Compact-DDM | $p$-value (unpaired T-test) |
|---|---|---|---|
| Group 1 (P5A0) | 5.23 (0.05) | 5.88 (0.12) | <0.01 |
| Group 2 (P0A5) | 4.46 (0.03) | 4.09 (0.02) | <0.01 |
| Group 3 (P2.5A2.5) | 4.77 (0.08) | 4.58 (0.03) | 0.009 |
| Group 4 (P0A0) | 4.71 (0.11) | 4.50 (0.01) | 0.012 |
| Group 5 (Biotene) | 13.63 (0.30) | 9.96 (0.03) | <0.01 |

The comparison of averaged viscosity (mPa·s) obtained from the rheometer and the compact DDM at the overlapped shear rate (6.38 to 17.65 $s^{-1}$). The $p$-value indicates the difference of value within the same row.

obtained from the rheometer for all materials ($p<0.05$). The highest difference was observed in the case of Biotene.

## Assessment of remineralizing actions

The highest median (min to max) of the percentage increase of the mineral-to-collagen ratio ($Abs_{1024}/Abs_{1636}$ ratio) at 7 days was observed with group 1 (22.95%, 6.09 to 63.83%), whilst the lowest value was observed with group 6 (3.54%, -27.82 to 10.74%) (Fig 4A). The value of group 1 was significantly higher than group 2 (-0.49, -34.65 to 6.89) ($p = 0.002$) and group 4 (-0.24%, -7.32 to 5.78%)($p = 0.008$). The value of group 3 (8.46%, 1.83 to 58.53%) was also significantly higher than that of group 2 ($p = 0.026$). The increase in the mineral-to-collagen ratio of Biotene was comparable to all experimental materials ($p>0.05$).

At 14 days, the increase in mineral-to-collagen ratio was detected in all groups except for group 6 (Fig 4B). The value of group 1 (25.48%, 6.03 to 52.65%) was significantly higher than group 6 (0.56%, -38.94 to 2.00%) ($p = 0.0002$). Group 3 (12.80%, 1.93 to 44.15%) also showed a higher mineral-to-collagen ratio than Group 6 ($p = 0.0236$). The value of group 2 was comparable to groups 4, 5, and 6 ($p>0.05$). A significant increase in the mineral-to-collagen ratio at 2 weeks was detected with groups 2 ($p = 0.008$) and 4 ($p = 0.002$).

## Assessment of surface mineral precipitation

The SEM image (Fig 5) of the specimen's surface showed that the surface of groups 1 and 3 exhibited the precipitates occluding dentinal tubules. The EDX results also demonstrated that

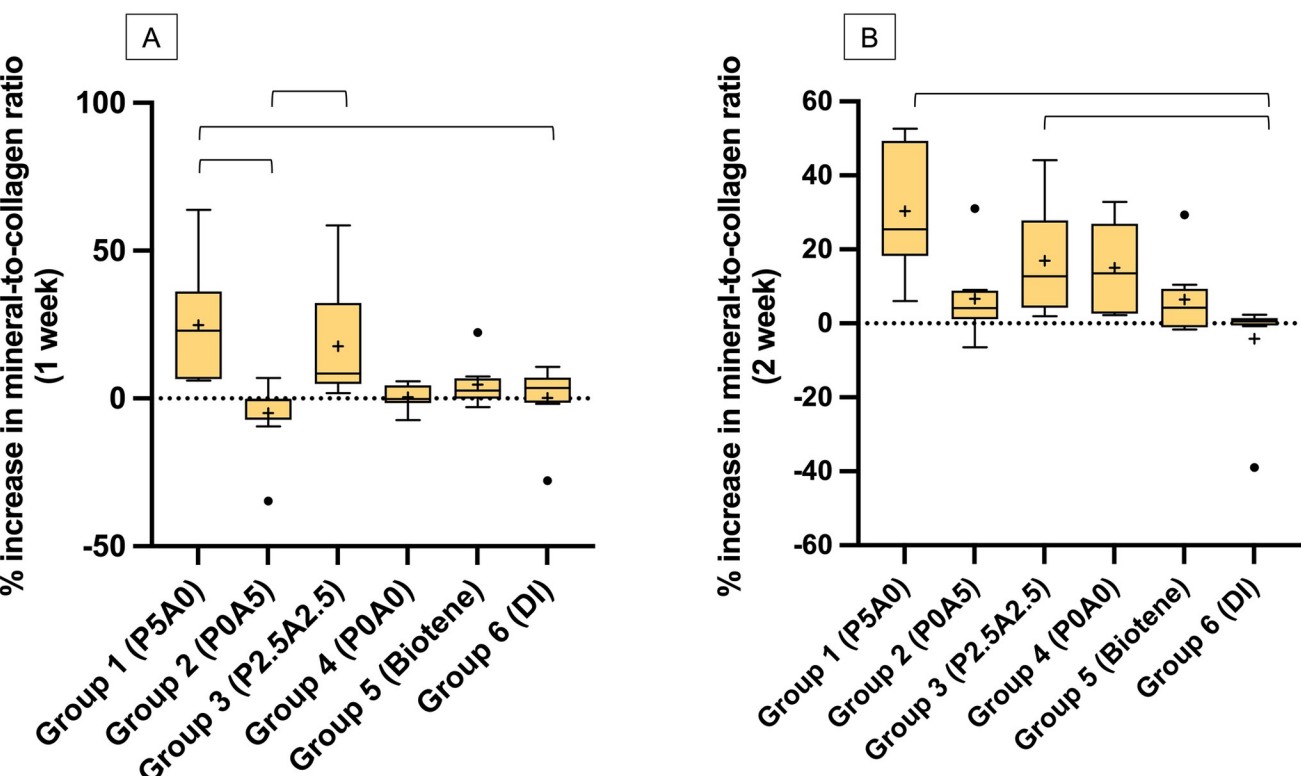

**Fig 4. The percentage increase of the mineral-to-collagen ratio ($Abs_{1024}/Abs_{1636}$ ratio).** The result from FTIR The percentage change of Abs1024/Abs1636 after pH cycling at (A) 1 week and (B) 2 weeks. The boxes represent the first quartile (Q1) to the third quartile (Q3). The horizontal lines in the boxes represent the median. The circles are outliers. The whiskers represent maximum and minimum values, and "+" represents the mean value (n = 8). The lines indicate $p<0.05$.

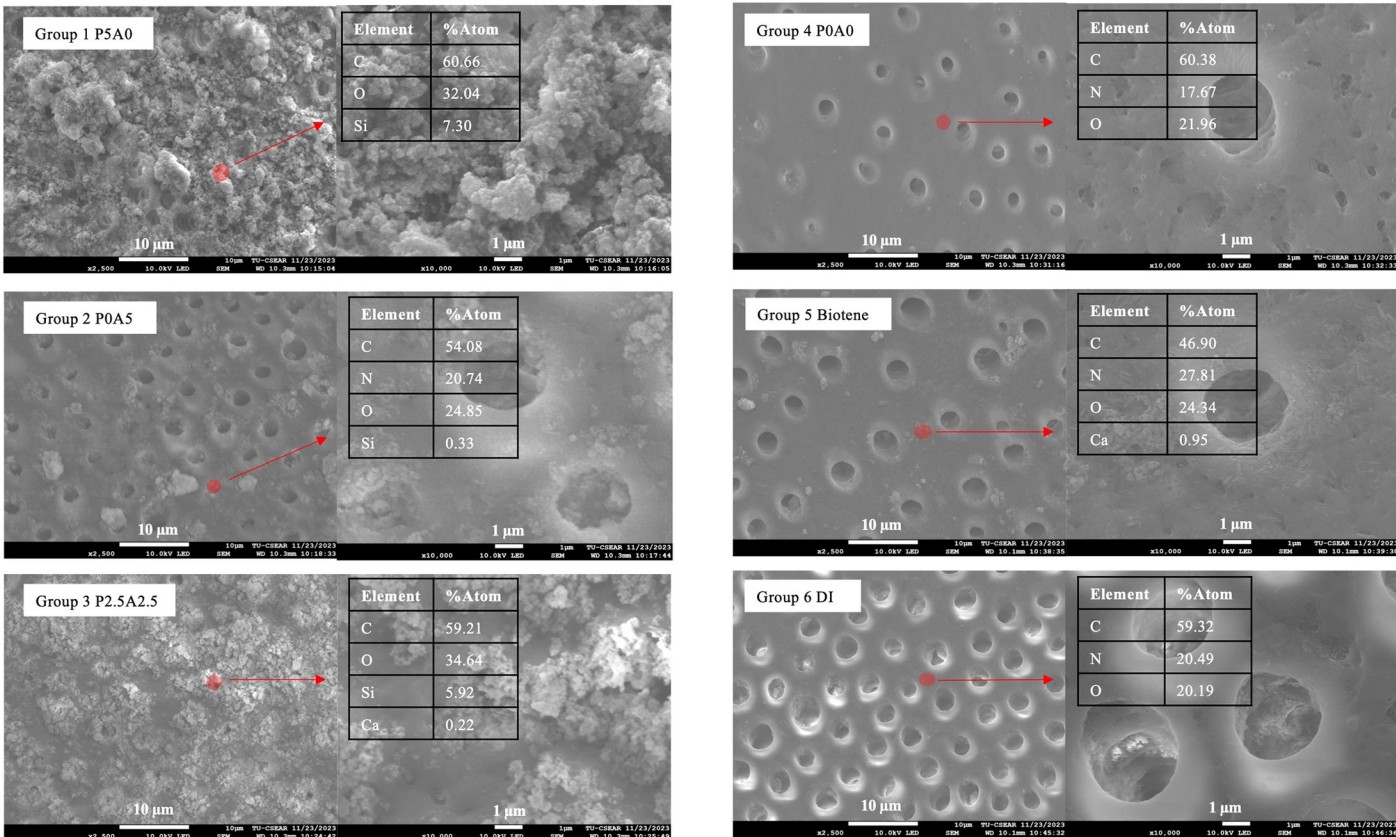

**Fig 5. SEM image of the representative sample after pH cycling for 14 days.** The precipitation and tubule occlusion were clearly observed in group 1 (P5A0) and group 3 (P2.5A2.5). Some areas of mineral deposition were detected in group 2 (P0A5), group 4 (P0A0), and group 5 (Biotene). In group 6 (DI), the tubule occlusion or mineral precipitation was not detected. Circles indicate the area of EDX analysis.

the elemental composition of the precipitate could be the silica-based compounds. The tubule occlusion was detected in some areas of groups 2, 3, and 5. No detection of tubule occlusion was observed in group 6 (DI).

## Discussion

It should be noted that the current study is an *in vitro* study, so the clinical relevance should be carefully interpreted. The results showed that the additives affected rheological properties and remineralizing properties. Hence, the null hypothesis was rejected.

The prepared saliva substitutes were expected to demonstrate rheological properties that closely mimic natural saliva, which varies with the fluid's shear rate. Different oral activities, such as swallowing and speaking (high shear rate of 60 to 160 $s^{-1}$) or resting (low shear rate of 0.1 to 1 $s^{-1}$), influence these shear rate [26]. The current study focused on shear rates from 0.1 to 100 $s^{-1}$, thus covering rest and active functions such as swallowing and speaking. It was reported that human saliva viscosity ranges from approximately 2.75 mPa·s to 15.51 mPa·s at a shear rate of 0.5 to 94.5 $s^{-1}$ [45]. Therefore, the viscosity of the experimental saliva substitute (4.71–5.24 mPa·s at a shear rate of 10–100 $s^{-1}$) still fell within the expected range of human saliva.

Shear thinning or the reduction of viscosity upon increasing shear rate is the natural characteristic of human saliva [46]. This property of structural fluid aids in forming a fluid film

that covers the food bolus for easier swallowing and in creating thin mucosal pellicles for the protection of lining mucosa in the oral cavity [12]. The current study found that none of the materials showed a clear reduction in viscosity with an increased shear rate or exhibited non-Newtonian behavior [47]. This was consistent with the previous studies, which reported slight non-Newtonian behavior in various commercial carboxymethylcellulose-based saliva substitutes [12, 26]. Further research can be, therefore, directed towards refining the formulation to enhance the rheological properties of the experimental saliva substitutes. It was suggested that incorporating mucin-based solutions such as bovine submandibular mucin or bovine serum albumin or alternative polymers like xanthan gum or scleroglucan could enhance the rheological properties to mimic human saliva [26, 46, 48]. Future studies should explore the use of these additives.

In this study, we also employed the innovative Compact Deep-Learning Assisted Platform for Micro-Rheological Assessment of Micro-Volume Fluids (compact-DDM) to assess the rheological properties of experimental saliva substitutes. This method, focusing on the analysis of particle flow in fluids, may provide a cost-effective and efficient alternative to traditional rheometers for assessing the rheological properties of oral fluids. The results indicated that viscosity measurements obtained using our compact-DDM were consistent with those from a strain-controlled rheometer, aligning with findings from a previous study [31]. However, a notable discrepancy was observed in the viscosity values of Biotene measured using the compact-DDM, which were lower compared to the rheometer readings.

We hypothesize that the high viscosity of Biotene might restrict particle movement, thereby impeding the accuracy of the compact-DDM measurements. This limitation is particularly significant since the compact-DDM's predictions rely mainly on thermal energy, with no external force applied to the system. To address this issue, smaller colloidal tracer particles could be employed to facilitate movement in a viscous medium. However, the size of the tracer must be large enough to be optically resolved by the compact-DDM, which requires a minimum size of approximately 0.35 μm. If the tracers are smaller than this, they cannot be detected in the acquired images, leading to background and noise signals predominating.

The discrepancies between the two methodologies for rheological assessment were detected in our study. This could be due to the different scales of measurement. The length scale measured using our compact-DDM (micro-rheology) is in micrometers, which is 1000 times smaller than the millimeter scale obtained by mechanical rheometers (macro-rheology) [49]. Consequently, the micro-rheology approach is more sensitive to weak responses compared to macro-rheology [50]. However, conventional mechanical rheometry remains preferable for characterizing solutions with high mass concentrations [51]. This suggests that compact-DDM might be more suited for analyzing fluids with lower viscosity. Additionally, the effective shear rate for compact-DDM's predictions, being below approximately $10–20\ s^{-1}$, may not be adequate for evaluating fluid behaviors during dynamic oral activities. The rheological tests were performed at room temperature to allow direct comparison due to the current limitations of compact-DDM. Future work should focus on improving the instrument to enable temperature control, which will allow the test to be performed at body temperature. In addition, replace the employed camera with a higher frame rate camera to increase the accessible shear rate range in our future work. However, the local rheological properties might differ from bulk rheological properties when the probe particle approaches the mesh microstructure and may appear hindered or caged, particularly in the rheological investigation of polymers [52]. Overcoming this limitation requires the bead radius to be considerably larger than a characteristic dimension of the material, such as the mesh size of a polymer network [52]. It should be noted that this method requires only a microscope, which could be practically useful in less-equipped settings [31].

The FTIR peak corresponding to phosphate group (~1024 cm$^{-1}$) of mineral apatite was shifted due probably to the alterations of the crystallinity of the apatite upon the demineralization/remineralization process [53, 54]. The previous *in vitro* study highlighted the role of mucin-like compounds such as hydroxyethyl cellulose [55]. These compounds help act as a protective barrier for the demineralization of apatite crystals. However, they may affect the mineral's reprecipitation by forming complex molecules with calcium, preventing ion exchange, and leading to unpredicted remineralizing outcomes [55]. This could be the explanation for the unclear mineral precipitation observed in groups 3,4, and 5.

The current study demonstrated the beneficial effects of propolis in enhancing dentin remineralization and tubule occlusion. This also correlates with the previous study, which indicated that artificial saliva mouth spray containing propolis reduced mineral loss from enamel [56]. The concentration at 2.5 or 5 wt% was chosen based on the ability of the extract to be mixed with saliva substitute from the pilot test. It should be mentioned that the exact mechanisms of propolis extract in remineralizing actions have not yet been fully established. The exact composition of the propolis extract was also not provided by the manufacturer. However, we hypothesize that propolis may promote remineralizing effects via three main mechanisms. Firstly, the resinous propolis extract may form a resinous film or seal on the dentin surface [57], which may prevent acid penetration and promote mineral deposition of the surface and the saturation of ions on the tooth surface. Furthermore, it was also reported that flavonoids, the commonly found phenolic substance in propolis, could reduce dentin demineralization and promote dentin remineralization via several mechanisms [58]. Phenolic hydroxyl groups may form hydrogen bonds with the amide carbonyl or hydroxyl group of collagen, thus enhancing the functionality of collagen fibrils that are essential for mineralization [59]. Additionally, flavonoids inhibited matrix metalloproteinases (MMPs), stabilizing collagen to mitigate acid penetration and the loss of calcium and phosphate from dentin [60, 61]. Future studies should confirm the effect of propolis on collagen degradation using hydroxyproline release assays or microscopic observation [62]. Furthermore, flavonoids may form complexes with calcium ions via their hydroxyl groups, facilitating the formation of nucleation sites necessary for the growth of mineral crystals [63]. The negatively charged non-collagenous proteins in the organic matrix of collagen contained highly charged phosphorylated serine and threonine residues that attract and trap calcium ions, encouraging the nucleation and growth of hydroxyapatite [64].

A concern with propolis extract is its potential to cause tooth staining due to its yellow color. This can have a negative impact on the aesthetic appearance, particularly with regular use. Additionally, phase separation of propolis in the solution was observed. The extracts in this study were sourced as food supplements, which may minimize the toxicity concern. However, the toxicity test should be evaluated to ensure its biocompatibility. This is especially crucial for vulnerable patients who have severe dry mouth symptoms or thin mucosa.

Based on the FTIR study, it was anticipated that groups 1 and 3 would exhibit higher levels of calcium and phosphorus detection in the SEM-EDX analysis compared to other materials. However, the EDX results did not show high levels of these elements on the surface of representative specimens. SEM results also indicated tubule occlusion effects in specimens immersed in the saliva substitute added with propolis. This occlusion could be crucial for providing a physical barrier to block dental tubules, potentially reducing dentin sensitivity [65, 66]. Nonetheless, the expected beneficial effects of aloe vera extract on dentin remineralization were not observed. The hypothesis was that the addition of the extract would lead to a reduction in pH, potentially resulting in conditions unfavorable for the mineral precipitation [14].

In the current study, a 1.1% fluoride gel (5,000 ppm) was used to simulate daily oral care at home in high-risk patients [67]. Despite using toothpaste with a high fluoride concentration,

the EDX analysis failed to detect fluoride on the surface of the representative specimens, which could be attributed to the small quantity of fluoride present. The presence of high silica content on the surface and the specimen's rough or irregular topography may additionally hamper fluoride detection [68]. Therefore, future studies should consider employing more sensitive detection methods such as X-ray photoelectron spectroscopy (XPS), X-ray diffraction (XRD), or PIGE (Particle-Induced Gamma-ray Emission) [69]. It is also hypothesized that fluoride may have formed as $CaF_2$ on the tooth surface [70], but it is possible that any formed $CaF_2$ was subsequently dissolved in the demineralization solution.

The current study offers saliva substitute formulations with added natural extracts that are simple to prepare, especially in oral health community centers with limited resources. The incorporation of natural extracts was expected to increase its appeal and could subsequently increase patient acceptance, particularly among those suffering from xerostomia [71, 72]. However, the study has certain limitations. For example, the direct comparison between different formulations was not feasible, as specimens in each group were not obtained from the same tooth. The large variation in the mineral-to-matrix ratio observed across formulations may be attributed to the inherent compositional differences of each tooth. In future studies, using specimens from the same tooth may yield more consistent results. However, this method may pose practical difficulties in handling small specimens. Furthermore, the demineralization/remineralization models employed chemical methods, such as EDTA and pH cycling, primarily due to their simplicity, low cost, efficiency, stability, and reproducibility [73]. It should be noted that the current study performed the pH cycling for 14 days which may not represent the long-term performance of the artificial saliva. Future work should consider employing a biofilm model with a longer study period, which would better simulate the biological and microbial aspects of oral environments and offer more clinically relevant insights into the caries process [73].

## Conclusion

The addition of propolis and aloe vera extracts increased the viscosity of the experimental saliva substitutes but showed minimal effect on viscoelastic properties. The experimental saliva substitutes showed viscosity lower than that of the commercial material but still within the range of natural saliva. The incorporation of propolis extract enhanced dentin remineralization and promoted the precipitation of minerals on the exposed dentinal tubules. These beneficial properties of the extract could potentially help promote the repair of dentin and reduce dentin sensitivity in xerostomia patients.

## Supporting information

**S1 File. Raw data of all experiments.** The raw data from rheological test and remineralization assessment of this study.
(ZIP)

## Acknowledgments

We would like to thank Thammasat University Research Unit in Dental and Bone Substitute Biomaterials, Thammasat University, Pathum Thani, Thailand, for supporting this study.

## Author Contributions

**Conceptualization:** Thanapong Intharah, Ungkarn Jarujareet, Piyaphong Panpisut.

**Data curation:** Surapong Srisomboon, Thanapong Intharah, Ungkarn Jarujareet, Arnit Toneluck, Piyaphong Panpisut.

**Formal analysis:** Surapong Srisomboon, Thanapong Intharah, Ungkarn Jarujareet, Arnit Toneluck, Piyaphong Panpisut.

**Funding acquisition:** Thanapong Intharah, Ungkarn Jarujareet, Piyaphong Panpisut.

**Investigation:** Surapong Srisomboon, Thanapong Intharah, Ungkarn Jarujareet, Arnit Toneluck, Piyaphong Panpisut.

**Methodology:** Surapong Srisomboon, Thanapong Intharah, Ungkarn Jarujareet, Arnit Toneluck, Piyaphong Panpisut.

**Project administration:** Thanapong Intharah, Ungkarn Jarujareet, Piyaphong Panpisut.

**Resources:** Thanapong Intharah, Ungkarn Jarujareet, Piyaphong Panpisut.

**Software:** Piyaphong Panpisut.

**Supervision:** Piyaphong Panpisut.

**Validation:** Thanapong Intharah, Ungkarn Jarujareet, Piyaphong Panpisut.

**Visualization:** Ungkarn Jarujareet, Piyaphong Panpisut.

**Writing – original draft:** Thanapong Intharah, Ungkarn Jarujareet, Piyaphong Panpisut.

**Writing – review & editing:** Thanapong Intharah, Ungkarn Jarujareet, Piyaphong Panpisut.

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
