## [Decision Letter · Decision Letter 0]

11 Apr 2024

PONE-D-24-02712The assessment of rheological properties and dentin remineralization of saliva substitutes containing propolis and aloe vera extractsPLOS ONE

Dear Dr. Panpisut,

Thank you for submitting your manuscript to PLOS ONE. After careful consideration, we feel that it has merit but does not fully meet PLOS ONE’s publication criteria as it currently stands. Therefore, we invite you to submit a revised version of the manuscript that addresses the points raised during the review process.

**ACADEMIC EDITOR: **

**I thank you for submitting an interesting manuscript. However, there are significant number of queries raised by the reviewers' which would make your manuscript more robust if corrected.**

We look forward to receiving your revised manuscript.

Kind regards,

Tanay Chaubal

Academic Editor

PLOS ONE

Journal Requirements:

"National Science, Research and Innovation Fund (NSRF) via the Program Management Unit for Human Resources and Institutional Development, Research and Innovation (Grant number B05F650016)"

Reviewers' comments:

Reviewer's Responses to Questions

**Comments to the Author**

1. Is the manuscript technically sound, and do the data support the conclusions?

Reviewer #1: Partly

Reviewer #2: Partly

2. Has the statistical analysis been performed appropriately and rigorously? 

Reviewer #1: Yes

Reviewer #2: Yes

3. Have the authors made all data underlying the findings in their manuscript fully available?

Reviewer #1: No

Reviewer #2: No

4. Is the manuscript presented in an intelligible fashion and written in standard English?

Reviewer #1: Yes

Reviewer #2: Yes

5. Review Comments to the Author

Reviewer #1: The article “The assessment of rheological properties and dentin remineralization of saliva substitutes containing propolis and aloe vera extracts” deals with a very interesting subject and would be a valuable contribution to literature. However, some critical issues should be addressed.

TITLE

The type of study (animal study, in vitro, or in vivo study) would be appreciated in the title of the article. Preclinical studies, such as in vitro trials, help us measure drug efficacy before administering a medicine to humans. The problem with them, however, is that there is a big difference between a drug administrated in vitro and the same drug administered in a human, so this information is important in the title and its clinical relevance should be carefully interpreted at the beginning.

ABSTRACT

The main conclusion of the article should be state on the abstract instead of “These promising properties could be beneficial for the prevention of dental caries in xerostomic patients” (line 38-39).

INTRODUCTION

The text is consistent; it contains sufficient and organized information about the topic to its purpose. However, some information should be addressed.

62-63: “The commercially available prescribed saliva substitute has limited specific therapeutic effects, such as mineralization and antimicrobial actions.” What are the available options? Reference studies that proved its limitations.

Furthermore, some given information is not referenced. For example: the sentence from line 51-52 and 80-81; Please add references, even if the authors are talking about the reference exposed at the end of the paragraph, after each phrase it should be exposed. Multiple in-text citations to the same work over a large section of text can be visually jarring and it is true that they are not entirely necessary. However, the rule of thumb is to cite the very first sentence, make it clear you are still talking about the same work in your subsequent sentences (for example, "The study noted that..."), and then confirm you are still talking about the work by including another citation at the end (if this has continued for several sentences).If you have a simple follow-on sentence in which it is still clear that you are talking about the same work, you do not need the reference in the second sentence. If at any point you think it might not be clear in the sentence that you are still referring to the same work, include another in-text citation, otherwise, it will appear as an information without a proper citation.

MATERIALS AND METHODS

96-97: The authors should at least summarize how the artificial saliva was performed in the text, even if described in another paper, you should briefly describe how it was performed in your article and then reference the source.

134-135: Again, explain how it was performed. What is the difference from previous studies? Why?

Overall, the information in this section is very condensed, even though you have to summarize the steps performed, all the information should be stated to ensure reproducibility.

RESULTS AND DISCUSSION

234: The aim of the study is already stated on the introduction section and it should not be here again.

Some statements also require references. The text also would benefit from a discussion with more studies addressed to discuss the findings of the present study.

Reviewer #2: I wanted to take a moment to express my appreciation for your diligent work on The assessment of rheological properties and dentin remineralization of saliva substitutes containing propolis and aloe vera extracts. Your efforts in conducting this research and contributing to the body of knowledge dental science are commendable.

I have the following comments:

1-The percentage mentioned, forecasting the increase in the proportion of elderly individuals from 12% to 22% by 2050, is derived from the WHO's Ageing and Health report, and is not attributed to AlQranei et al. (2020) [1]. For accurate sourcing, please refer to the following link: https://www.who.int/news-room/fact-sheets/detail/ageing-and-health. Accessed on March 24, 2020.

2-The prevalence of dental caries, particularly root caries, among the elderly is undoubtedly a significant oral health concern. However, to ensure the currency and robustness of the information presented, it is imperative to incorporate more recent statistics or references regarding this issue. Additionally, it's crucial to recognize that the percentage cited, originating from Lopez et al. (2017), is based on a review article primarily focused on studies conducted in the USA and other developed countries. Consequently, there may be limitations in extrapolating these findings to developing countries, where there could be potential variations across different demographics or geographical regions.

3-The chemical formula and mechanism of action of propolis and aloe vera extracts are essential aspects to elucidate in your manuscript. Could you kindly include detailed explanations of their chemical compositions and mechanisms of action? This information will enhance the understanding of their therapeutic potential and contribute to the comprehensiveness of your study.

4-While the introduction of your manuscript provides an overview of the techniques utilized to assess rheological characteristics, such as strain-controlled rheometer and compact-DDM method, it lacks an exploration of potential limitations or challenges associated with these techniques. For instance, it would be beneficial to address concerns regarding the accuracy, consistency, and comparability of measurements obtained using the compact-DDM method, particularly in laboratory settings. Please consider further examination of these aspects to provide a comprehensive understanding of the methodology employed in your study.

5-It is essential to acknowledge the potential variability in individual responses to experimental saliva substitutes, which may not have been fully addressed in the introduction of your manuscript. Considering the diverse responses among individuals, it would be beneficial to discuss this aspect further to provide a comprehensive understanding of the study's implications.

6-kindly include group 6 distilled water in Table 2 of your manuscript? This addition will ensure completeness and clarity in presenting the experimental groups and their respective treatments.

7-Could you please provide clarification on why the sample temperature was maintained at 25°C rather than at 37°C, which is the normal temperature in the oral cavity? This deviation from physiological conditions warrants explanation to ensure the relevance and applicability of your experimental setup to real-clinical scenarios.

8-Could you kindly explain the rationale behind choosing to cut the crown of the teeth horizontally instead of perpendicularly? Given that a perpendicular cut would likely result in a greater exposure of dentinal tubules compared to a horizontal cut, I am interested in understanding the reasoning behind this decision. Your insights would be greatly appreciated.

9-Could you kindly include more details in your manuscript regarding the aperture size used during ATR-FTIR measurements, the surface treatment and polishing of the samples, as well as the dehydration process and rehydration?

10-I kindly inquire whether the phenomenon of crack formation in the samples due to the process of dehydration and rehydration has been observed in your study. Understanding any potential effects of these processes on sample integrity would contribute to the robustness of your findings.

11-I kindly note that the graph indicates a significant shift in the phosphate peak, which appears to be unreported. Additionally, in such cases, it is preferable to measure the area under the curve rather than focusing solely on peak height for a more comprehensive analysis.

Moreover, there are other important parameters worth considering to better understand the effects of propolis and aloe vera. These include the carbonate to phosphate ratio, collagen cross-linking, and the degree of crystallinity of dentin. It is evident from the chart that these parameters are also influenced to a significant extent. Integrating these aspects into your analysis would enhance the depth of understanding of your study outcomes.

12-I would like to kindly point out that the study employs a simplified demineralization model involving EDTA. While this approach serves its purpose, it may not fully replicate the complex and multifactorial nature of natural caries development. Considering the diverse factors involved in caries formation, such as biofilm presence, incorporating more diverse demineralization models could enhance the clinical relevance of the remineralization test.

13-I kindly note that the remineralization protocol in your study spans only 14 days, providing valuable insights into short-term effects. However, it may be beneficial to consider extending the duration of the study to assess the sustainability and long-term remineralization potential of the saliva substitutes. Longer-term studies could offer valuable insights into the durability of remineralization outcomes over extended periods, thereby enriching the comprehensiveness of your findings.

14-Could you kindly provide information regarding the acquisition parameters used to collect the EDX data, including the kilovoltage (kV) and working distance (WD)? Additionally, I recommend calculating the ratios of calcium to phosphorus (Ca:P) and calcium to carbon (Ca:C) to further elucidate the elemental composition. Moreover, I emphasize the critical importance of thorough data analysis from SEM images in dentin samples. Considering the varied composition of different dentin areas, such as intertubular dentin and peritubular dentin, careful analysis is essential to ensure a comprehensive understanding of the findings. It would be helpful to specify from exactly where the measurements were taken to provide clarity and context to the results.

15-I kindly note that while the results of your study suggest promising trends in terms of rheological properties and remineralization potential, it is important to consider several limitations. Specifically, the discrepancies observed in viscosity measurements between the rheometer and compact-DDM highlight potential technical challenges or limitations in accurately assessing rheological properties, particularly at higher shear rates. Acknowledging and addressing these limitations will strengthen the interpretation and applicability of your findings.

16-I kindly suggest discussing the results of remineralization in more depth, considering the chemical mechanism of remineralization of demineralized dentin. Exploring the mechanisms underlying the observed effects, including the role of specific components in propolis and aloe vera extracts, would provide valuable insights for clinical applications. Your attention to these aspects would enrich the discussion and enhance the relevance of your findings for clinical practice.

6. PLOS authors have the option to publish the peer review history of their article (what does this mean?). If published, this will include your full peer review and any attached files.

Reviewer #1: No

Reviewer #2: **Yes: **Hamza Elfarraj

---

## [Author Response · Author response to Decision Letter 0]

19 Apr 2024

Please see the attached letter of respond to reviewer.

---

## [Decision Letter · Decision Letter 1]

8 May 2024

The in vitro assessment of rheological properties and dentin remineralization of saliva substitutes containing propolis and aloe vera extracts

PONE-D-24-02712R1

Dear Dr. Piyaphong Panpisut, 

We’re pleased to inform you that your manuscript has been judged scientifically suitable for publication and will be formally accepted for publication once it meets all outstanding technical requirements.

Kind regards,

Tanay Chaubal

Academic Editor

PLOS ONE

Additional Editor Comments (optional):

Reviewers' comments:

Reviewer's Responses to Questions

**Comments to the Author**

1. If the authors have adequately addressed your comments raised in a previous round of review and you feel that this manuscript is now acceptable for publication, you may indicate that here to bypass the “Comments to the Author” section, enter your conflict of interest statement in the “Confidential to Editor” section, and submit your "Accept" recommendation.

Reviewer #1: All comments have been addressed

Reviewer #2: All comments have been addressed

2. Is the manuscript technically sound, and do the data support the conclusions?

Reviewer #1: Yes

Reviewer #2: Yes

3. Has the statistical analysis been performed appropriately and rigorously? 

Reviewer #1: Yes

Reviewer #2: Yes

4. Have the authors made all data underlying the findings in their manuscript fully available?

Reviewer #1: Yes

Reviewer #2: No

5. Is the manuscript presented in an intelligible fashion and written in standard English?

Reviewer #1: Yes

Reviewer #2: (No Response)

6. Review Comments to the Author

Reviewer #1: The authors addressed all the comments appropriately, making the article clearer and more consistent to be published, in my opinion. Thank you for effort to make this happen.

Reviewer #2: I would like to extend my honest appreciation for the extraordinary quality of your work. The scientific rigor observed in your manuscript is commendable - it offers invaluable insights into how rheological properties and dentin remineralization of saliva substitutes containing propolis and aloe vera extracts can be assessed. What sets this paper apart is not just its depth of information but also the clarity with which that information has been presented.

It is clear from start to finish that you have devoted a lot of time into doing thorough research as well as analyzing data collected during such studies; this dedication shines through every page. Moreover, what you have written is easy to follow thanks largely in part to good organization on your part.

In conclusion then, kudos should go out towards such an outstanding piece of work that contributes greatly towards this particular area of study. The fact alone that it is also very informative besides being scientifically sound means there’s much knowledge we can gain from what you’ve done so far; thus eagerly anticipate any future input generated as per your ongoing research efforts.

Thanks again for a job well done…

7. PLOS authors have the option to publish the peer review history of their article (what does this mean?). If published, this will include your full peer review and any attached files.

Reviewer #1: No

Reviewer #2: No

---

## [Editor Report · Acceptance letter]

10 May 2024

PONE-D-24-02712R1 

PLOS ONE

Dear Dr. Panpisut, 

I'm pleased to inform you that your manuscript has been deemed suitable for publication in PLOS ONE. Congratulations! Your manuscript is now being handed over to our production team.

Kind regards, 

on behalf of

Dr. Tanay Chaubal 

Academic Editor

PLOS ONE